# Effect of a Comprehensive Health Care Program on Blood Pressure, Blood Glucose, Body Composition, and Depression in Older Adults Living Alone: A Quasi-Experimental Pretest–Posttest Study

**DOI:** 10.3390/ijerph17010220

**Published:** 2019-12-27

**Authors:** Eun Jeong Hwang, In Ok Sim

**Affiliations:** 1Department of Nursing, Sehan University, Jeollanam-do 58447, Korea; ejhwang@sehan.ac.kr; 2Red Cross College of Nursing, Chung-Ang University, Seoul 06974, Korea

**Keywords:** older adults, blood pressure, blood glucose, depression, senior center, comprehensive health care

## Abstract

This study explored the effects of a comprehensive health-care program (CHCP) on blood pressure, blood glucose, body composition, and depression in older adults living alone. We used a quasi-experimental, two-group, pretest–posttest design. The CHCP consisted of open lectures, health counseling, exercise classes, nutrition counseling, and self-help group meetings at a local senior welfare center. Fifty-eight subjects participated in this study, with thirty subjects in the experimental group and twenty-eight subjects in the control group. Data were analyzed by using the descriptive statistics, *χ*^2^-test, and *t*-test. Comparisons of the pretest and posttest systolic blood pressure (*t* = −2.530, *p* < 0.016) and blood glucose (*t* = 3.089, *p* < 0.004) between the experimental and control groups showed significant differences. In both the experimental (*t* = 3.949, *p* < 0.001) and control groups (*t* = 3.816, *p* < 0.002), depression symptoms showed a significant decrease posttest, compared with pretest. Our findings infer that older adults require physical and psychosocial health care and that more efforts must be made to improve the general health and well-being of this population group.

## 1. Introduction

In 2015, older adults—members of the population 65 years and older—accounted for 12.8% of the total population. This number increased to more than 14% in 2018 [1]. The number of older adults living alone is also continuously increasing due to changes in the social environment. This number increased from 13.6% in 1994 to 21.2% in 2017, and it is expected to increase to 23.3% by 2035 [1]. 

Older adults living alone may experience social isolation, deterioration of social relationships, and hampered development [2,3]. The main difficulties faced by older adults are economic and health related, and these difficulties may include feelings of loneliness and alienation; those who are living alone may experience these problems more intensely [4]. In a survey of older adults living alone, conducted by the Ministry of Health and Welfare [2], 73.3% of the respondents replied that they had contact with their family less than once every month, while 57.3% of respondents said that they had almost no contact with others within their neighborhoods, and 50.0% reported not participating in any social activities. Older adults living alone have reported significantly lower levels of psychological well-being, self-esteem, and life satisfaction than older adults living with others [5]. Cederbom et al. [6] state that living alone is related to poor health status, difficulties with activities of daily living, low levels of physical activity, and a greater prevalence of chronic pain. Furthermore, older adults living alone are more likely to have poorer physical and mental health statuses, as they tend not to adopt health-promoting behaviors, such as exercise, health care, nutrition, and stress management [3]. Poor physical and mental health in all older adults may lead to serious social problems; however, these problems are likely to be even more serious in older adults living alone. Another notable factor is, as age increases, the risk of being left alone due to the death of a spouse increases; therefore, the likelihood of becoming lonely and alienated also increases. In South Korea, the suicide rate among older adults is 104.5 per 100,000, which is more than double the total suicide rate in 2012 [7].

Among older adults living alone, the severity of their physical and mental problems often depends on their level of social support. Zebhauser et al. [8] report that older adults with a stable social network are less likely to feel depressed or lonely. The absence of depression and a functioning social network are considered to be the factors most likely to protect older adults living alone from loneliness. When older adults living alone have these social relationships, they are able to maintain communication with the outside world and overcome the feelings of redundancy that accompanies old age, as well as the depression caused by the loss of family members [9]. However, those that do not receive adequate help or support from family or others are at risk of depression, which exacerbates mental and physical health issues and negatively affects their quality of life. It has been reported that a significant negative relationship exists between loneliness and overall social support [9].

Several related studies have shown that older adults living alone require more social support, stronger relationships, and better physical health than older adults living with others. Current programs addressing the health and well-being of older adults living alone are limited to interventions that focus on disease management and an integrated approach. These programs are often inadequate for meeting the requirements of this population. Chronic disease requires continuous care throughout life, to improve and maintain self-function and to provide effective education, which enable individuals to control their illnesses. Senior welfare centers are well used by older adults, and therefore, it seems reasonable that integrated programs that focus on chronic-illness management and social-relationship formation should be administered at these centers [10]. Senior welfare centers currently provide various programs for older adults that involve hobbies, leisure, employment and health support, and general education; however, it has been reported that existing emotional and social support programs are insufficient [10]. The fragmented nature of these programs, however, makes it difficult to address the physical and mental health and emotional problems of older adults. It has previously been advocated that social contexts and health-related factors should be considered in an integrated manner, to resolve difficulties such as depression, loneliness, and the many other problems faced by older adults living alone [11]. This suggests that a comprehensive program involving health education, counseling, and self-help groups may be appropriate. 

This study explored the effects of a comprehensive health care program (CHCP) on blood pressure, blood glucose, body composition, and depression in older adults living alone in M city. Ultimately, this study aims at providing basic data to improve CHCPs at senior welfare centers, for older adults living alone.

## 2. Materials and Methods

### 2.1. Design

The study was conducted by using a quasi-experimental pretest–posttest design; participants were divided into experimental and control groups.

### 2.2. Participants

The subjects of this study were adults aged 65 years and older who are living alone, who had been diagnosed with hypertension or diabetes, and who lived near and could walk to a senior welfare center in M city. Participants were recruited by using various methods, including an announcement on the welfare center homepage and a notice board, the distribution of promotional materials, and recommendations from a community health center. The minimum sample size required for a two-tailed *t*-test analysis was determined by using G*power software (version 3.0)(Faul, F.; Erdfelder, E.; Lang, A.-G.; Buchner, Olshausenstr. 40, D-24098 Kiel, Germany) and setting the effect size at 0.8, the significance level (α) at 0.05, and the power (1 − *β*) at 80%. The minimum sample size required was calculated to be 52 individuals, 26 in each study group [12]. Sixty-two subjects agreed to participate in the study and stated that they understood the purpose of the study. Of these, thirty-two subjects who agreed to participate in the CHCP were assigned to the experimental group. The remaining thirty subjects chose not to participate in the program but agreed to data collection and constituted the control group. Members of the control group participated in a recreation program run at the welfare center. During this intervention, two subjects in each group withdrew from the experiment due to the move and health deteriorating. A final sample of fifty-eight subjects participated in the study: thirty subjects in the experimental group and twenty-eight subjects in the control group.

### 2.3. Data Collection

#### 2.3.1. Blood Pressure

The same qualified and experienced nurse measured all the subjects’ blood pressure, using an upper-arm cuff blood pressure monitor (BRAUN ExactFitTM 5 upper arm blood pressure monitor, Kaz Europe Sàrl, Lausanne, Switzerland). Normal adult systolic and diastolic blood pressure readings were set to be 120 and 80 mmHg, respectively [13]. 

#### 2.3.2. Blood Glucose 

Blood glucose levels were measured by the same nurse at the same time, using a finger-prick blood glucose monitor (SD CodeFreeTM 5 blood glucose monitoring system, SD Biosensor Co., Ltd., Chungbuk, Korea). The normal range for blood glucose was set as 80–130 mg/dL in a fasting state and less than 180 mg/dL two hours after a meal [14]. 

#### 2.3.3. Body Composition

Body composition consists of muscle mass (MM) and body fat mass (BFM), and both were measured by using a body component meter (InBody 4.0 unit, Biospace Co., Seoul, Korea), in the standing position, with bare feet positioned as indicated on the unit. The arms grasp the electrode knobs to complete the machine’s measurements.

#### 2.3.4. Geriatric Depression Scale: Short Form

We used the Korean version of the geriatric depression scale: short form—originally developed by Yesavage and Sheikh [15] and modified by Kee [16]—to assess depression. This scale consists of 15 items with “yes” or “no” answers. Scores range from 0 to 15 points, and a score of 4 points or below indicates a normal result, a score of 5 to 9 points indicates mild depression, and 10 points and higher indicates severe depression. The Cronbach’s ⍺ of this instrument in the present study was 0.88, indicating acceptable internal consistency. 

#### 2.3.5. The CHCP Process

The CHCP applied in this study was drafted based on a comprehensive literature review, after which the first draft was revised and confirmed by consulting with community experts, doctors, nursing professors, and local experts. Finally, a six-month CHCP was finalized and used in this study. A nurse at the welfare center coordinated the practice team—consisting of nurses, physiotherapists, and nutritionists—and held monthly meetings to verify progress. The internists who provided the open lectures were commissioned from the community. Details concerning the CHCP are provided in Figure 1.

The CHCP was implemented for six months, from February to July 2017. It consisted of open lectures, health classes, exercise classes, nutrition classes, and self-help group meetings. The program contents were as follows: open lectures, one each, based on three themes—dementia and depression, hypertension, and diabetes mellitus—were presented by local internists. These lectures lasted for approximately one hour each. Every time the participants visited the center, their blood pressure and blood glucose levels were checked by the same nurse, at the same time. At the health classes, sixty one-hour self-disease-control education classes and one-on-one health counseling sessions were provided by the same nurse at the center, every Monday and Wednesday. Sixty one-hour exercise classes were provided by a physical therapist, every Tuesday and Thursday. These consisted of exercises focused on strength, balance, agility, and flexibility. The same exercise program was provided every time, to all participants, by the same instructor. At the nutrition classes, sixty one-hour nutrition counseling sessions were provided by a nutritionist, every Monday and Friday. The topics included diet menus, nutrition education, and counseling. Two self-help group meetings were held at which subjects shared experiences and coping strategies. Lectures and counseling sessions were held in a seminar room and a consultation room, respectively. The exercise programs were conducted in an auditorium. A free lunch was provided daily for any of the older adults that visited the center. This contributed to the subjects rarely being absent, despite the six-month study period. To ensure the validity of the study results, the researchers and the program’s service providers were not in communication. The researcher that analyzed the results did not participate in the program, and a different experienced nurse than the nurse who was involved in the program collected data from subjects before and after the program. This nurse was unaware of the participants’ group assignment. To prevent the effects of the program affecting members of the control group, intervention times differed and classrooms were some distance from each other. 

### 2.4. Data Analysis

Data were analyzed by using IBM SPSS version 21 software (IBM Corporation, Armonk, NY 10504, USA). The homogeneity comparison of the general characteristics between the experimental and control groups was used as the chi-squared test and independent sample *t*-test. Comparisons between the experimental group’s and control group’s blood pressure, blood glucose levels, body compositions, and depression levels before and after the CHCP were performed by using a paired *t*-test and independent sample *t*-test. 

### 2.5. Ethical Considerations

Before conducting this study, approval was obtained from the institutional review board of the clinical examination committee of Sehan University (SH-IRB 2018-25). Participants were provided with an explanation of the study purposed and informed that they were free to withdraw from the study at any time, without prejudice. Written informed consent was obtained from each individual. Identification data of study subjects were kept separately at the welfare center and were not made available to those conducting the study. The CHCP for the experimental group and a play program for the control group were provided free of charge.

## 3. Results

### 3.1. Participant Characteristics

The general characteristics of the experimental and control groups are shown in Table 1. There were no significant differences concerning gender, age, or disease status between the two groups. There were 11 males (36.7%) and 19 females (63.3%) in the experimental group and 17 males (60.7%) and 11 females (39.3%) in the control group. The mean age was 78.17 (±4.92) years old for the experimental group and 77.64 (±6.45) years old for the control group. In the experimental group, 18 subjects (60.0%) had hypertension, two subjects (6.7%) had diabetes, and ten subjects (33.3%) had both diseases. In the control group, 14 subjects (50.0%) had hypertension, three subjects (10.7%) had diabetes, and 11 subjects (39.3%) had both.

### 3.2. Comparison of Blood Pressure, Blood Glucose, Body Composition, and Depression before and after Participating in the CHCP

The results of the blood pressure readings, blood glucose levels, body composition readings, and levels of depression of the experimental and control groups’ pretest and posttest are shown in Table 2.

The pretest results show no significant differences in blood pressure, blood sugar, body composition, and depression between the experimental and control groups. The systolic blood pressure and blood glucose levels of the experimental group showed significant changes in the posttest readings, compared with the pretest readings; however, the systolic blood pressure (*t* = −2.91, *p* < 0.008), blood glucose (*t* = −3.68, *p* < 0.001), and muscle mass (*t* = 2.13, *p* = 0.046) of the control group was significantly changed posttest, compared with the pretest. The systolic blood pressure (*t* = −2.53, *p* < 0.016) and blood glucose (*t* = −3.09, *p* < 0.004) differences between posttest and pretest between the experimental and control groups were significant. In both the experimental (*t* = 3.95, *p* < 0.001) and control groups (*t* = 3.82, *p* < 0.002), depression was significantly decreased posttest, compared with the pretest results. Although not statistically significant, the muscle mass of the control group had generally decreased after the intervention, while the muscle mass of the experimental group had increased after the intervention.

## 4. Discussion

This study explored the effects of a CHCP on blood pressure, blood glucose, body composition, and depression in Korean older adults living alone. The results showed that, after the intervention, the overall systolic blood pressure in the experimental group decreased, compared with that of the control group. Moreover, the blood glucose levels in the control group were significantly higher than those in the experimental group after the intervention. In related studies [8,10,17,18,19,20], older adults that participated in a physical activity program showed physical function improvement, compared with older adults that did not participate. 

In this study, a slight increase in muscle mass was noted in the experimental group, whereas a significant decrease was observed in the control group (*t* = 2.13, *p* = 0.046). Existing studies show that limited physical activity in older adults causes muscle atrophy, which, in turn, reduces overall physical function [21,22]. These studies also suggest that physical function in older adults cannot be improved by temporary short-term physical programs, but that it can be improved by consistent long-term physical activity.

In this study, both the experimental group (*t* = 3.95, *p* < 0.001) and the control group (*t* = 3.82, *p* = 0.002) showed a significant decrease in post-intervention depression; however, the mean decrease in depression was greater in the experimental group (differences were −1.80 and −1.06, respectively). Furthermore, members of both the experimental and control groups—both of which took part in regular activities—showed improvements in depression, which is consistent with the results of previous studies [4,23]. The results of these previous studies support the notion that taking part in different activities improved depression in older adults. In their work, Cho and Lee [24] emphasize the importance of forming various social connections by performing community activities and establishing links with welfare services, based on their finding that greater social support improves life satisfaction. The deterioration of family function and the weakening of social support systems are thought to contribute toward exacerbating loneliness and depression and decreasing the quality of life in older adults. Kim and Kim [10] conclude that programs provided by senior welfare centers play an important role in the formation of social support systems.

The physical, mental, and social difficulties faced by older adults have increased along with the increased prevalence of older adults living alone. These individuals experience depression due to loneliness and isolation, which negatively affects their life satisfaction [23]. Lee and Kim [25] show that social support reduces stress, and therefore depression. Zebhauser et al. [8] conclude that older adults with a stable social network are less likely to feel depressed or lonely. Their work confirms that the absence of depression and the presence of a functioning social network are the most important protective factors against loneliness in older adults living alone. Furthermore, income, level of education, and age-related limitations were shown to have little impact. According to Kim [4], older women living alone experience positive interaction when participating in various group therapy activities, which positively affects their self-esteem, resulting in positive physical and psychological effects. According to Kim [23], nutrition and dietary habits (*β* = 0.42, *p* < 0.001) and regular activities (*β* = 0.37, *p* < 0.001) have a significant effect on the life satisfaction of older women living alone in rural areas. 

According to Lee and Kim [26], many older adults living alone are no enthusiastic about their lives, have no confidence in their health statuses, and do not feel like proud members of society. Dorji et al. [17] state that many older adults complained of depression. They also report that the physical and psychological conditions and social relationships of older adults were significantly correlated with their quality of care, and state that a variety of participation programs and health programs for older adults living alone should be provided to address this problem. 

Along with the rapidly increasing number of older adults, the proportion of older adults living alone continues to increase. Older adults living alone have been reported to have substantially poorer physical and psychological health. These factors and the pressure generated by changing demographics mean that there is an urgent need to determine the social alienation risks of older adults living alone and to reach a consensus on both the social support required and on means of promoting relationship formation.

## 5. Conclusions

This study aimed to determine the effects of a CHCP conducted at a senior welfare center for older adults living alone, considering their blood pressure, blood glucose, body composition, and depression. Due to population aging, the physical and psychosocial health of the older adults are becoming ever more important; furthermore, the increase in the proportion of older adults living alone compounds the problem. Those living alone are more vulnerable to a lack of social support than those living with others. Therefore, organizations for the support of older adults, such as senior welfare centers, should focus more on older adults living alone. The results of this study emphasize the need for a CHCP for older adults living alone; however, the provision of comprehensive community health-care programs is currently very limited. Based on our findings, we recommend that participatory programs be supported and revitalized by targeting welfare centers.

We also suggest that further research be conducted on the development of CHCPs by considering individual characteristics and circumstances, such as family support and future management plans.

Several limitations to this study warrant consideration. First, the results of this study may differ from the general situation, because only older adults wanted to participate in the program. Second, individual characteristics—such as disease severity or degree of family support—were not considered; therefore, caution should be used with the generalization and interpretation of our results.

## Figures and Tables

**Figure 1 ijerph-17-00220-f001:**
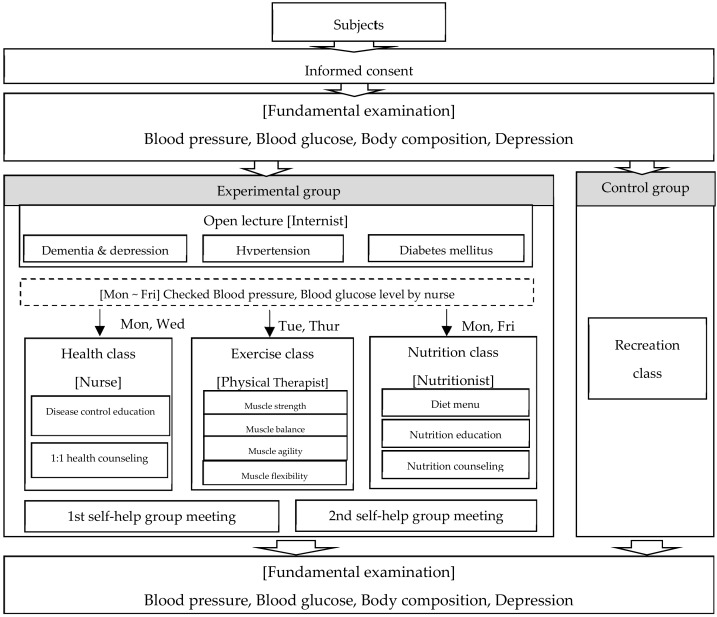
The comprehensive health-care program (CHCP) process.

**Table 1 ijerph-17-00220-t001:** Homeostasis of the general characteristics of the experimental group and the control group.

Variables	Experimental Group (*n* = 30)		Control Group (*n* = 28)		*t* or χ^2^	*p*-Value
	*n*	%	*n*	%		
**Gender**						
Male	11	36.7	17	60.7	3.35	0.058
Female	19	63.3	11	39.3		
**Age**						
65–69	1	3.3	3	10.7	1.30	0.521
70–79	15	50.0	12	42.9		
≤80	14	46.7	13	46.4		
M ± SD	78.17 ± 4.92		77.64 ± 6.45		0.35	0.728
**Disease**						
HT	18	60.0	14	50.0	0.68	0.712
DM	2	6.7	3	10.7		
Both	10	33.3	11	39.3		

HT = hypertension; DM = diabetes mellitus; BP = blood pressure.

**Table 2 ijerph-17-00220-t002:** Comparison of pretest and posttest blood pressure (BP), blood glucose (BG), body composition (BC), and depression.

Variables			Pretest	Posttest	t (*p*)	Differences	t (*p*)
BP	SBP	Experimental group	139.91 ± 21.65	132.96 ± 12.83	1.26 (0.222)	−6.96 ± 26.56	**−2.53 (0.016)**
		Control group	130.71 ± 14.98	139.88 ± 15.37	**−2.91 (0.008)**	9.17 ± 15.43	
		t (*p*)	1.69 (0.099)	−1.67 (0.102)			
	DBP	Experimental group	78.67 ± 17.42	81.47 ± 11.64	−0.65 (0.520)	2.800 ± 23.55	0.42 (0.674)
		Control group	76.81 ± 13.27	77.44 ± 13.45	−0.23 (0.824)	0.63 ± 14.53	
		t (*p*)	0.45 (0.656)	1.21 (0.231)			
BG		Experimental group	132.04 ± 36.17	131.35 ± 39.47	0.12 (0.904)	−0.69 ± 28.93	**−3.09 (0.004)**
		Control group	126.50 ± 39.75	161.71 ± 59.47	**−3.68 (0.001)**	29.30 ± 37.79	
		t (*p*)	0.52 (0.608)	**−2.37 (0.022)**			
BC	MM	Experimental group	21.59 ± 4.64	21.75 ± 4.72	−0.47 (0.644)	0.16 ± 1.87	1.44 (0.156)
		Control group	22.51 ± 4.20	22.03 ± 3.94	**2.13 (0.046)**	−0.48 ± 1.05	
		t (*p*)	0.69 (0.494)	−0.23 (0.820)			
	BFM	Experimental group	31.49 ± 8.92	32.34 ± 8.77	−0.87 (0.395)	0.85 ± 5.12	0.43 (0.670)
		Control group	32.59 ± 8.32	32.90 ± 7.83	−0.72 (0.479)	0.31 ± 1.82	
		t (*p*)	−0.30 (0.766)	−0.02 (0.987)			
Depression		Experimental group	7.60 ± 4.00	5.80 ± 2.93	**3.95 (0.000)**	−1.80 ± 2.50	−1.15 (0.255)
		Control group	7.47 ± 4.75	6.41 ± 4.00	**3.82 (0.002)**	−1.06 ± 1.14	
		t (*p*)	0.10 (0.921)	−0.60 (0.550)			

SBP = systolic blood pressure; DBP = diastolic blood pressure; MM = muscle mass; BFM = body fat mass. Note. The bold numbers in this table indicated significant values.

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
