# Peer review of "Effect of a Comprehensive Health Care Program on Blood Pressure, Blood Glucose, Body Composition, and Depression in Older Adults Living Alone: A Quasi-Experimental Pretest–Posttest Study"

_ijerph, 2019, doi:10.3390/ijerph17010220_

Round 1

Reviewer 1 Report

Manuscript Number: ijerph-673389

Full Title: Effect of a Comprehensive Health Care Program on Blood Pressure, Blood Glucose, Body Composition, and Depression in Older Adults Living Alone: A Quasi-experimental Pre-posttest Study

I am pleased to have the opportunity to review the study because the number of older adults living alone has also been continuously increasing that increases social risks in various parts. Overall, the structure and process of the research seem to have been well developed according to scientific research procedures. However, the authors expect to improve the quality of the research by reviewing and supplementing the following points. Details are as follows:

Page 2, L51. Please clarify “currently”.

Page 2, L85. It would be beneficial to describe the aim of the study including dependent variables at the end of the ‘Introduction’.

Page 2, L89. Please consider revising the Design section ‘The study was conducted using a quasi-experimental, two-group, pre-posttest design.’ Especially, the “two-group” may need other expressions.

Page 3, L103.  It is important to explore the reason for participants’ withdrawal from experimental studies. Please illuminate the reason in detail.

Page 3, L129. Please replace “Data processing” by other phrases such as “The CHCP program”. It would be helpful to illustrate the program using a table or a figure.

Page 5, L184. Please consider revising the title of ‘Table 1’. Especially, “Homeostasis” is used instead of “Comparison”.

Page 7, L260. It would be appropriate if descriptions regarding the ‘limitation of the study’ moved at the end of the “Discussion” section.

Author Response

Thank you for your kind and professional review.

I tried my best to fix it.

Thank you again.

Reviewer 2 Report

Thank you for the opportunity to review your paper.

This is an important area. Your findings support the plentiful global evidence regarding this age group, they do not provide any novel evidence.

I have made comments throughout your paper.

Good luck in your research endeavours.

Author Response

(The authors gave the same response as above.)
